# Users' Perceptions of Key Blockchain Features in Games

Iikka Paajala [1,*], Jesse Nyyssölä [1], Juho Mattila [1,2] and Pasi Karppinen [1]

1  Faculty of Information Technology and Electrical Engineering, University of Oulu, Pentti Kaiteran katu 1, 90570 Oulu, Finland
2  Ikune Labs, Loukkutie 11 A1, 90540 Oulu, Finland
*  Correspondence: iikka.paajala@oulu.fi; Tel.: +35-84-4010-1802

**Abstract:** The blockchain is an emerging technology that has the potential to revolutionize the gaming industry among a wide range of different business fields. So far, only a few studies have been conducted about blockchain gaming. This study introduces a mobile game utilizing blockchain asset tokens and smart contracts. It was developed for research purposes and used to demonstrate blockchain-based games using semi-structured interviews. This study follows the exploratory research paradigm, which aims to map research of little-known areas. This study focuses on how participants perceived blockchain attributes such as trust, transparency, and user-generated content and how this affected engagement and their willingness to play the game again. Based on our evaluation, generating blockchain assets positively impacted player retention. According to the results, providing genuine asset ownership through the blockchain contributes to environmental engagement and self-engagement, as well as player retention. Another positive blockchain feature discovered from the interview data is user-generated content implementation into games.

**Keywords:** blockchain; games; non-fungible token; qualitative study

## 1. Introduction

Blockchain is an emerging technology that has the potential to revolutionize the gaming industry among a wide range of different business fields [1–3]. Information systems research about the blockchain has remained scarce, and the need for more academic studies has been notable in recent years [3,4].

Blockchain games have not yet been able to leave a mark in the gaming industry; nevertheless, the fusion of games and blockchain shows immense potential [5,6]. So far, only a very few studies have been conducted about the usefulness of the blockchain in gaming [3]. The most well-known blockchain game Cryptokitties has been studied at least by Min et al. [3] and Jiang & Liu [7]. A systematic literature review on blockchain-based applications by Casino et al. [8] and Ali et al. [1] did not address blockchain games at all. Only a few years ago, the majority of blockchain studies were about Bitcoin [4]. Nowadays, research has also been done in sectors such as finance, government, manufacturing, and health [1,8]. Blockchain is still a relatively novel technology, and research on other topics will gradually emerge [2], and the expectations of blockchain will increase the demand for blockchain applications [9].

The use of the blockchain has several benefits [10]. For example, trust and transparency are argued to be inherent blockchain features [11]. In the gaming context, transparent blockchain data enables the players or third-party stakeholders to audit the smart contract and related game rules, enhancing the game's trustworthiness [3]. With the use of blockchain token technology, users generating their own content for games can be granted ownership and compensation via smart contracts [10]. This can encourage players to participate in content creation in several ways [3,10].

Player retention is crucial to the success of a mobile game [12] since not all mobile gamers return to the game after trying it shortly after its installation [13]. Player engage-

ment in game analytics aims to measure a user's behavior related to a particular game. Bouvier et al. [14] described engagement in terms of attention, immersion, involvement, presence, and flow.

This study introduces a mobile game, IkuneRacers, utilizing blockchain asset tokens and smart contracts. The game was developed for research purposes at the University of Oulu. This IT artifact was used as demonstration material when conducting semi-structured interviews. Based on the data analysis, key themes and categories were recognized. The aim of this study is to evaluate trust, transparency, and user-generated content through artifact demonstration and interviews. Furthermore, we examine whether blockchain implementation can influence player retention and engagement.

The research questions are:

RQ1: How do players perceive transparency, trust, and user-generated content in blockchain games?

RQ2: Does blockchain game implementation influence users' perception of their engagement and player retention?

The structure of the paper is as follows. First, we introduce blockchain research, how blockchain assets are related to games, and studies on player retention and engagement. After that, we explain how our blockchain game (demonstration material) was designed, the research methods, and how the data was collected. The results are presented in Chapter 4. Chapter 5 includes a discussion and draws conclusions, and presents future research opportunities and limitations of the study.

The presented data were collected for a master's thesis by one of the authors.

## 2. Background

Thus far, the blockchain has been primarily studied in practitioner literature in computer science and cryptography studies, but theory and empirically driven IS research on the blockchain is gradually emerging [15]. Rossi et al. [15] assess existing research on blockchains and provide a framework for blockchain research in information systems (IS). In their framework, the authors [15] provide research agendas for several fields in IS, from which 'Agenda for behavioral IS research on the blockchain' is the most relevant for this study. Rossi et al. [15] call for more studies on the blockchain to be done either on the protocol level, application level, or in between. On an application level, one crucial issue is blockchain adoption [15]. Although several promising blockchain applications have been proposed, widespread adoption is rare [2]. Rossi et al. [15] believe privacy, security, and scalability are key factors, and research on these topics could make both academic and practical contributions [15].

### 2.1. Blockchain Assets and Games

A blockchain game can offer the potential of true ownership of crypto assets, as the asset ownership can be stored with a smart contract to the blockchain. Also, smart contracts enable the transfer of assets without third-party involvement [16]. Pillai, Biswas, and Muthukkumarasamy [17] define crypto assets as digital assets that are stored on the blockchain and utilize various techniques (e.g., cryptography, distributed consensus, peer-to-peer network, smart contract) to create, transact, and verify in a decentralized manner. They presented three categories of crypto assets: asset-tokens, crypto-coins, and utility-tokens. In their framework, the assets can be classified based on the fungibility and tangibility of the asset [17]. Table 1 illustrates the characteristics of different digital tokens, and Table 2 further explains them. In the context of the blockchain, a token is tangible if it represents something with a tangible existence, such as company shares or digital art. Tangible tokens can also represent ownership, e.g., in a physical world like property or a car [17]. If the asset is unique, it cannot be traded directly into similar tokens (aka a non-fungible token, NFT). An asset is fungible if it is exchangeable using a common standard, value, and characteristics like crypto coins (i.e., cryptocurrencies such as Bitcoin and Ether), which are fungible and intangible assets used as a means of exchange. Lastly,

utility tokens are intangible and non-fungible and represent access to a product or service and also serve paying functions [17,18]. One example of utility tokens is the ERC-20 token standard used in the Ethereum network [18]. Asset-tokens can be fungible or non-fungible. Pillai, Biswas, and Muthukkumarasamy [17] use the CryptoKitties game as an example of tangible and non-fungible asset-tokens. It has similar principles to what was used when designing this game. In Cryptokitties, players "breed" digital cats 'CryptoKitties', which are non-fungible tokens (ERC-721 token standard), as each is unique [19].

**Table 1.** Types of digital assets [17].

|  | **Tangible** | **Intangible** |
|---|---|---|
| Non-Fungible | Asset-token | Utility-token |
|  | ERC-21 token | |
|  | passport | Service |
| Fungible | Asset-token | Cryptocoins |
|  | ERC-20-token | Bitcoin |
|  | Car | Ether |

**Table 2.** Further explanation of crypto assets. Table modified from Pillai, Biswas and Muthukkumarasamy [17].

| | **Crypto Coins** | **Crypto Assets Asset Tokens** | **Utility Tokens** |
|---|---|---|---|
| Fungibility | Fungible | Fungible or non-fungible | Non-fungible |
| Tangibility | Intangible | Tangible | Intangible |
| Represents | Digital object, medium of exchange | Object with tangible characteristic | Digital object providing access rights |
| Example | Bitcoin, ETH | Car, property, NFT | Service, subscription |

Min et al. [3] categorized blockchain-based games and divided them into four groups based on their utilized blockchain characteristics: rule transparency, asset ownership, (cross-platform) asset reusability, and user-created content. Curran [5] added to that list decentralized asset ownership and exchange, fast and secure payment networks, and the ability for developers to monetize their creations. The most popular blockchain-based benefits for games are rule transparency (i.e., gambling) and asset ownership (i.e., collecting in-game items) [5]. Standardized cross-platform asset reusability could benefit games utilizing asset ownership [5]. The ownership of an individual crypto asset can be validated using smart contract standards, which adds trust to the system and therefore gives the users confidence to invest in it [5].

### 2.2. Blockchain Trust, Transparency, and User-Generated Content

According to Queiroz and Wamba [11], both trust and transparency are built-in blockchain features. In fact, blockchain systems tend to be safer and more transparent than regular databases [8]. In the blockchain environment, transparency also covers asset trading [16]. As blockchain-based public ledgers are transparent, trust in the system increases [10,11]. In the gaming environment, this allows the players or third-party stakeholders to audit the smart contract and related game rules [3]. A further aspect of the blockchain, the reduced need for intermediaries, can also benefit the developers as they can build and distribute their products directly to the customer via smart contracts [5].

User-generated content can be used to create and update game elements, which can be crucial in retaining veteran players [3]. However, this topic can include built-in development opportunities for users, especially in blockchain game environments, and the created material can even be preserved and shared among multiple games. Furthermore, blockchain can store the ownership information of user-generated material. Blockchain and smart contracts have also been proposed as a method for paying for contributions to intellectual property [10]. These characteristics suggest that blockchain implementation can encourage players to participate in content creation in several ways [3,10]. As an example,

in the game Last Trip, the developers provide story frames whereby the players contribute content to the game [3].

Although blockchain technology enables co-content creation by developers and non-developers alike, this study concentrates on asset ownership within the possibilities of blockchain technology. Asset ownership was implemented on the application on the first (and current) iteration.

### 2.3. Retention and Engagement

Predicting player retention is crucial to the success of a mobile game [12]. The reason for this is that a considerable number of mobile gamers do not return to the game after trying it shortly after its installation [13]. Player retention can be measured in numerous ways, one being the 'Retention profile,' which provides valuable information for the developers on how different user segments and cohorts play the game and how long they will keep playing [20]. Drachen et al. [12] analyzed variables that affect player retention in a free-to-play mobile game with a dataset of over 130,000 players. The key findings of the study were that the most positive predictors of retention included variables that tracked long and consistent participation. An interesting finding by Drachen et al. [12] was that good results in the early stages resulted in lower retention, which suggests that the players found the games too easy and lost interest. However, according to Weber et al. [21], games should not be too hard to play either. The blockchain-related issue of the gas fee has been proven to affect player retention [22], but this paper does not attempt to study that phenomenon further.

Bouvier et al. [14] described engagement with the terms: attention, immersion, involvement, presence, and flow. When a player's attention is towards game content, but the consciousness is in the real world, the state is engagement. When the player's consciousness shifts into the game world, the player feels a presence, and flow is an action-oriented aspect of presence [14]. In addition, gamers eventually abandon the game without proper motivation [23]. Motivation can be divided into intrinsic (playing the game itself) and extrinsic motivations (e.g., rewards and punishments) [24]. Concentrating on intrinsic reasons, motivation can be further divided into human needs, such as competence, autonomy, and relatedness [24]. These needs are positively linked to game enjoyment. The need for competence in games is met through challenges and goals. The need for autonomy is met if the player is given the power to make choices during the gameplay. Relatedness can be satisfied if games offer social or online interactions [24]. In this study, we are not measuring flow, although we do ask for interviewees' opinions on the matter.

### 3. Research Methods

To find answers to research questions involving such novel technology and topics as blockchain and asset tokens, we build an artifact to demonstrate a blockchain-based game and tokens to support the interviews. The interviewees got familiar with the game and the selected blockchain mechanics prior to the interviews.

This study follows the exploratory research paradigm (e.g., [25], which is described as mapping research of little-known areas of study. It aims to discover the nature of the problem and guide academics to understand it better. Exploratory research is flexible and can use both quantitative and qualitative methods. However, exploratory research does not aim to test a hypothesis based on existing theories and knowledge.

### 3.1. The Artifact

The artifact developed in this study is a blockchain-based mobile multiplayer game prototype. The game is a turn-based racing game where players tap their cars for them to move forward. The game demonstrates blockchain features, as the cars, including their attributes, such as name, color, and speed, are blockchain items. Additionally, the user accounts are managed via smart contracts. In the current version of the game, players receive a prize in the form of coins for participating in races, which can generate more cars.

Figure 1 illustrates the garage screen view of the designed game. Racer names are generated randomly and work as a seed for the blockchain to calculate attributes for the racer object. The column "Dice" shows the speed value of the car, which consists of three dice-related attributes, which are represented in the format "xDy + z." Here, x refers to the number of dice thrown, whereas D is short for dice, y represents the number of die sides, and z is a constant modifier added to each result. In addition to the dice values, the blockchain stores several other values, like the racer's level, its model and color, the number of wins, and the races it has participated in. The model and color of the racers are values stored in the blockchain, but apart from other attributes, they are represented visually, as shown in a table format on the garage page (Figure 1).

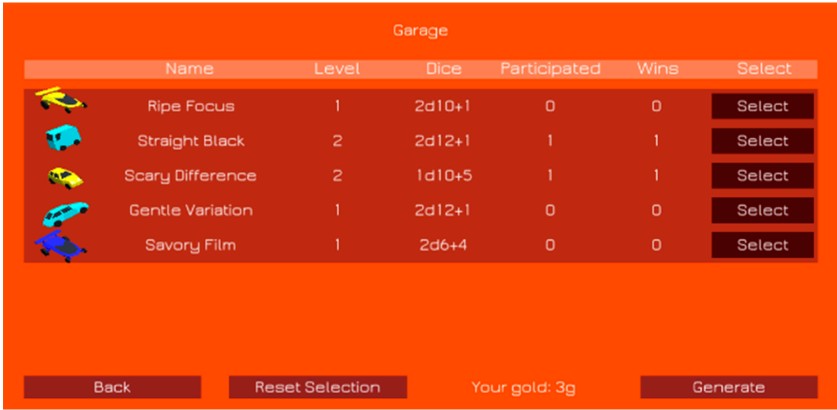

**Figure 1.** Garage view of the game.

For blockchain implementation, the platform used was Loom, which is meant for decentralized applications and includes a software development kit (SDK). For development, testing, and data gathering purposes, we used the testnet provided by Loom called Extdev instead of the Ethereum or the Loom mainnet. Extdev is free for development purposes and has the same functionality as the mainnet, including a web platform that enables examining the generated blocks. Both the Ethereum and Loom mainnets require tokens that have real-world value.

The smart contracts of the application were written in a Solidity language and built and deployed with Truffle. Smart contracts define the way the data is stored in the blockchain, and the functions used to manage the blockchain and applications built on it. Visual Studio Code (text editor) with a Solidity extension was used for development, while Visual Studio 2019 was chosen for Unity development. Loom's SDK enabled implementing queries to the blockchain as if they were built locally. This made it possible to retrieve relevant data from the blockchain.

As indicated by Rossi et al. [15], designing smart contracts for the artifact plays a key role in blockchain implementation. The smart contracts followed many of the principles by Tonelli et al. [26] to make them modular and service-oriented. The smart contract functionality in the game in this study was based on well-defined autonomous tasks and was, therefore, authority-free, which means that there were no privileges or exclusive rights written in the code of the blockchain interactions. To minimize the risks of smart contract-related logical errors and software bugs, it was decided to have only the racer and account data stored and managed on the blockchain while everything else is handled by the client. Well-defined and autonomous smart contract tasks also help to reduce complexity and therefore reduce the mentioned risks.

The account is identified based on the private key. When creating an account, the "GenerateKeys()" function is called. Based on this key, the client will use a hash function to calculate its address, which the blockchain will use to identify the user. "GetAccountName()" is called without parameters as the contract knows who is calling based on the address.

### 3.2. Demonstrations and Interviews

Ten game demonstrations, along with semi-structured interview sessions, were conducted either face-to-face or online due to the COVID-19 pandemic in the spring of 2020. During the online meetings, the participants could not try the artifact themselves but had to rely on a thorough gameplay demonstration through screen sharing. Two of the interviews were held in English, and the other eight were in Finnish. Some of the interviewees were recruited based on their participation in a blockchain course at the university, while others were recruited through personal connections.

All the participants were studying or graduated from the University of Oulu or Oulu University of Applied Sciences. Three of the interviewees were studying in or had a degree from a university of applied sciences, and the other seven were studying in or had a degree from a university.

Because the blockchain is a novel technology and could be unfamiliar to the general population, a demonstration of the artifact and explanation relevant to blockchain-related qualities took place prior to the interviews. This is supported by Schultze and Avital [27], as the purpose is not to affect interviewee responses but to hand out more perspectives for interviews. The artifact was demonstrated to 10 participants before being interviewed. The demonstration included a hands-on gameplay session (face-to-face) or a gameplay showcase (online), open discussion on the game and blockchain, and showcasing the generated blocks on the web platform. All the interviews were conducted, recorded, and coded by the second author.

After the game demonstration, the conducted interview consisted of questions focused on the topics presented in the Background section. These include how interview participants regard blockchain-related aspects (such as asset ownership, asset value, immutability, transparency, security, and trust) and how those are linked to video game characteristics like retention and engagement.

The interview question structure contained the following five topics:

1. Background questions
2. What makes assets valuable
3. Asset ownership and the lack of a trusted third party
4. Data on the blockchain is immutable and transparent
5. Security and trust of blockchain systems

The interviews were semi-structured and utilized laddering, and the questions depended on those elements the interviewees found personally meaningful. Interviews followed the method by Schultze and Avital [27] to collect relevant data for this study. For interview perspectives, Schultze and Avital [27] take up a combination of localistic and romantic views. The localistic view allows the interview to be a platform for activities, i.e., political action or impression management, which is affected by different contexts, such as the interviewee's age or gender. In a way, interviews are seen as windows into social reality [27]. In the romantic view, interviews are conversations in which the interviewer can participate and intervene when necessary [27]. Naturally, the interviewer is not fishing for preferred answers but handing out more perspectives [28]. Also, according to Holsten and Gubrium [28], interviews are not only for stating facts but also for constructing meanings.

Schultze and Avital [27] suggest three scientific interviewing methods that can be used for IS research. These are appreciative, laddering, and photo-diary interviewing. For this study, the laddering interviewing method was selected. In practice, questions such as "How do X and Y differ from Z?" are used to make distinctions between elements. On the other hand, questions like "Why is Z important?" aims to create a meaningful connection between elements [27]. Here, the interviews aimed to identify the meaningful connections between video games and blockchain features.

Because blockchain is a novel technology and could be unfamiliar to the general population, a demonstration of artifacts and explanations of relevant blockchain-related qualities took place prior to the interviews. This is supported by Schultze and Avital [27],

as the purpose is not to affect interviewee responses but to hand out more perspectives for interviews.

After demonstrating the game, the conducted interview consisted of questions focused on the topics presented in the Background section. These include how interview participants regard blockchain-related aspects (such as asset ownership, asset value, immutability, transparency, security, and trust) and how those are linked to video game characteristics like retention and engagement.

The interview question structure contained the following five topics:

1.  Background questions
2.  What makes assets valuable
3.  Asset ownership and the lack of trusted third party
4.  Data on blockchain being immutable and transparent
5.  Security and trust of blockchain systems

The interviews were semi-structured and utilized laddering, and the questions depended on those elements the interviewees found personally meaningful. Table 3 presents the interview method and language used in interviews. The interview questions are listed by the topic in the Appendix A.

**Table 3.** The interview method and language of the interviewee.

| Interviewee Number | Interview Method | Interview Language |
| :---: | :---: | :---: |
| #1 | Face-to-face meeting | English |
| #2 | Face-to-face meeting | English |
| #3 | Online meeting | Finnish |
| #4 | Face-to-face meeting | Finnish |
| #5 | Online meeting | Finnish |
| #6 | Online meeting | Finnish |
| #7 | Online meeting | Finnish |
| #8 | Online meeting | Finnish |
| #9 | Online meeting | Finnish |
| #10 | Online meeting | Finnish |

### 3.3. Data Analysis

The data analysis followed a thematic analysis suggested by Terry et al. [29], and it consisted of six steps. The first step is familiarization, wherein some insights are gained from the data. Here the entire dataset is gone through, notes are taken, and some early analytic ideas are formed. [29] In the second step, the data is reduced and organized into patterns. The data mass becomes an easily organizable set of labels, which are generated based on the relevancy of the research questions. This might indicate that some datasets have several labels and some none [29].

Theme development is a process of pattern identification, and that happens in the third step of the thematic analysis [29]. A central idea or concept supporting a set of codes is developed for a theme. According to Terry et al. [29], in this step, visual aids, such as tables and figures, are crucial to demonstrating different themes and relationships between them, as well as identifying new themes.

In the fourth step, the themes are reviewed so that they work with the labeled data, datasets, and research questions. The fifth step is the phase for defining the theme, whereby the researcher interprets and describes each theme and its representation. Also, "thin" themes are dropped out. The sixth and final step is reporting [29].

### 4. Results

The research results reported here focus on blockchain issues of transparency, trust, user-generated content, player retention, and engagement. Also, some gameplay (involving the blockchain) improvement suggestions were gathered from the interviews, including a leveling system wherein the players can improve and customize their garages and

cars. The interviewees were also interested in generating blockchain objects and were hoping to see more content on that. In the following sections, the participants' (P1–P10) responses are divided into themes in accordance with the research questions. Some of the participants' quotations include comments in parentheses that represent the interviewer's interpretations. Interview topics and identified themes and subthemes are presented in Table 4. The interview data is from the master's thesis of Jesse Nyyssölä [30].

**Table 4.** Interview topics and corresponding themes with sub-themes.

| Interview Topic | Themes | Sub-Themes |
|---|---|---|
| General questions | Personal relationship with video games | Personal motivation |
| | | Emotions in video games |
| | Blockchain-related experience | Cryptocurrencies |
| | | Blockchain games |
| | Blockchain-related experience | Suggestions for improvement |
| | | Retention on artefact |
| | | Engagement on artefact |
| What makes assets valuable | What gives value to an asset? | Assets with monetary value |
| | | Assets with non-monetary value |
| | Giving all assets monetary value (i.e., making them sellable and purchasable) | Financial incentive |
| | | Unfair playing field in video games |
| Asset ownership and the lack of a trusted third party | Authorities over data in a video game context | Authorities controlling/ exploiting |
| | | High prerequisites for authority-free environment |
| | Exchange through a central system as opposed to person-to-person | Convenience of centralized system |
| | | Social aspect of person-to-person exchange |
| | Actual ownership of virtual assets | |
| Data on the blockchain, immutable and transparent | The effects of transparency in video games and IS | Traceability and accountability |
| | | Privacy concerns |
| | Third-party involvement in video games | Continued support |
| | | Modding |
| Security and trust on the blockchain | Trust on the blockchain | |

### 4.1. Retention, Engagement, and Gaming Experience

General questions included basic questions about the artifact and gaming and background information about the participant. For the question, 'Why do participants choose to play video games over other forms of media entertainment,' the major answer included mentions of active participation in games (nearly all participants) or social aspects of games (P3, P7, P10). Answers also included topics such as 'Autonomy over your character' (P4, P6, P8) and the 'Variety of environments and interacting with it,' while one interviewee focused on the role of thinking and decision-making as the following quotation reveals:

*"Video games are great because you get to use your brain, create things yourself, decide on the course of the events yourself, how it is going, what you are allowed to do, what you want to do. In streaming services, you can choose what to watch."* (P8)

Playing video games can be relaxing and reduce stress by giving something else to think about (P8). Also, the feeling of immersion was mentioned (P3). Most participants noted that playing video games involves a wide range of emotions, including excitement, annoyance, nostalgia, and fearfulness:

*"If you are looking for alleviating stress, it (the emotion) can be, for example, calmness. For example, in some role-playing games like Witcher, you can see yourself running in the virtual world and hear the wind in your ears, and some music can calm the situation at the same time. It is so versatile in terms of which emotions and senses touch you."* (P6)

The interviewees who stated their interest in video games knew little or nothing about blockchains. Some of them knew about Bitcoin or other cryptocurrencies, but the

participants who were interested in the blockchain had more varied background knowledge on the topic. For games in general, they suggested a few blockchain-related characteristics, such as the private key (P5), currency conversion (P6), asset ownership (P9), and asset reusability (P3). Specifically for the artifact, seven participants suggested improvements, including more customization, visual aids, and more refined progression and competition systems.

> *"I like the idea of cross-platform or cross-game related things very much. [ . . . ] If we imagine that you have a Mario game and they would release four different games where you could get this cross-stuff from each game so that you could shuffle them around with this blockchain-like idea, I would probably buy them all. [29]"* (P3)

One participant (P2) stated that the game was not interesting, but the blockchain implementation seemed interesting from a technical point of view. Another participant (P3) also noted that the genre of the game is not relevant, but the uniqueness of the virtual assets provides an exciting setting. Another participant saw the game just as a technical demonstration but, at the same time, supported the core ideas of the blockchain functionality:

> *"I would play a game like that, but of course not with cars because I think cars are the most boring thing. However, if we were to say that something like Diablo would work that way, that all items are blockchain objects, it would be a really good thing. Having items in a game be reflected in the outer world or other currency, I do not see that it would make any game worse. [29]"* (P8)

Four participants stated that they might want to play again (P2, P4, P6, P10). This was largely attributed to the interest in generating new cars. However, two participants (P2, P10) said that they would only play it a few times because the game lacked meaningful gameplay content.

When asked about the perceived engagement of the artifact, the answers varied. One noted that the current game is not engaging, but the fundamental idea of a car race could be (P9). Two participants noted that the competitive spirit of a car race was an engaging factor (P4, P6). Two participants were keen on generating tokens (P5, P8). For one participant, the engagement was enabled by the possibility of progressing to better vehicles (P7). Re-playability (retention) and engagement by participants are shown in Table 5.

**Table 5.** Summary of the answers regarding re-playability and engagement.

|  | Play Again | Engaged |
|---|---|---|
| Yes | 4 | 2 |
| No | 3 | 2 |
| No answer | 3 | 1 |
| Partly | - | 5 |

To ensure that the participants could assess the effects of the blockchain regarding assets, it was necessary to establish how the participants perceived the value of their assets in general. All the interviewees gave an example of a valuable item in a video game, and some could distinguish between an item with monetary value and a non-monetary value. One participant gave an example of both types of valuable items:

> *"In Player Unknown's Battlegrounds, I had things that could be converted by selling to Steam currency, and that could be used to buy any games sold on Steam. Additionally, in games like the World of Warcraft, if you have spent a couple of years trying to find a certain item that you have a chance to get only once a week with a drop rate lower than one percent, so even if it is not in any way significant or it does not have monetary value, it still feels good to achieve it."* (P5)

When asked if the instrumental value of the game item could be made exchangeable with the value of money, some respondents feared that (at its worst) one would be able to

buy a competitive advantage. The topic was complicated since there are positive ideas of combining real money with games, like being able to sell your items after quitting the game (financial incentive), but also negative aspects like bots and possibly an unfair playing field. The most supportive comment was the following:

> *"It would not bother me even if we could trade genuinely valuable things. That would be fun. I can hear how my money goes down the drain. On the other hand, if everything was for sale, it could mean that using so-called 'honest methods,' it would be really difficult to get those items. That could be one negative drawback affecting people playing the game."*(P3)

One interviewee considered not spending money on video games to be a part of the challenge of the game:

> *"The worst thing you can do is that you buy with money the best item you can get. The game is over at that point. So, you are paying money so that you do not want to play the game anymore."* (P8)

Regarding the preference of trading person-to-person versus through a centralized system, most participants agreed that the option with the central system is better because it is often more convenient. However, there were also respondents (P6, P8) who advocated for person-to-person trading:

> *"Old RuneScape did not have Grand Exchange yet, . . . so people would gather in these trade worlds where thousands of people were in one place shouting what they wanted to sell. Some people wanted money in return, but some traded items for other items. It was a really functional and extremely pleasant social event where people were gathering as if they were at a marketplace."* (P6)

*4.2. Transparency, Trust, and User-Generated Content*

Transparency of video game data was one of our interview foci. This chapter presents findings regarding blockchain transparency, trust, and user-generated content. Transparency is related to the themes of traceability, accountability, and privacy concerns. One participant (P1) underlined the distinction between transparency and public data:

> *"I think transparency is important for everything. I have been contemplating public and private data for some time. While I believe that even if it is private, it should be transparent, not all private data can be public. Transparent means that you can see a log of the actions, and [so can] the public, [ . . . ] everybody can also see the data."* (P1)

According to one interviewee (P9), transparency can improve traceability and reduce application abuse. Five participants (P2–5, P6 & P10) pondered privacy issues.

> *"Surely it (transparent data in a system) is not better than private. If I happen to sell something, I do not want everybody to check that 'Oh, they sold that.' [ . . . ] If it does not enable everyone to see something that a regular guy would like to keep hidden, then it is not necessarily a bad thing."* (P10)

The interviewee's comments on trust in the blockchain were mixed. None of the participants could give a direct "yes" or "no" answer to the question, "From what you have experienced, would you say that you trust the blockchain?" Only three participants indicated at least some trust in the blockchain, but other comments were conditional, for example: "

If it is made in the right hands" or "depends on the implementation." Answers show suspicious attitudes towards novel technology participants are not yet familiar with. One participant clearly states reluctance to adopt blockchain in the first wave:

> *"I would say that with my understanding still, I would not (trust blockchain). [ . . . ] I have played games that are based on servers, and they have worked for me and everyone else, so I trust those. Maybe for me to trust the blockchain, it would require that it becomes*

*a mainstream thing and is proven safe. [ . . . ] I am not the first guy who goes to try a new invention."* (P10)

Closely related to immutability and transparency, one of the common features in blockchain applications is user-generated content. Since the game designed for this study did not incorporate user-generated content directly into the mechanics of the game, the interviews evaluated more external forms of user-generated content. Specifically, modifications and developing the game further after the original developers quit (i.e., preserving the game). All except one participant (P8) had positive reactions to these ideas.

*"The law of supply and demand comes straight away and an old saying that you should not shoot a milk-producing cow. What I mean by that is that when the developers stop developing the game, it is because no one wants to fund it anymore, no one wants to buy it, and it does not have any players."* (P8)

However, another interviewee (P6) had quite a different opinion on the matter:

*"That (open third-party involvement) would have saved so many good game projects so far. For example, Age of Empires Online, some eight years ago, was a concrete example that there was a game that was playable but ran out of developers. There were players, sure, but no one wanted to update and continue developing the game."* (P6)

According to one respondent (P4), the benefit of third-party modding is that mods often fix bugs that the developers have not been bothered to fix. Another participant (P6) states that mods enable playing older games with newer technology. From a business point of view, the relationship between the developer and the community can profit from modding (P5). One participant saw external modification as a failure of the developers:

*"I am just a gamer who does not want to use my own thinking time to ponder these modding issues, and I just play what is fun. [ . . . ] It is the developers' responsibility to get the people to remain in the game and to direct us players to do what makes us stay with the game."* (P8)

Although the questions regarding user-generated content within a game were not in focus, one participant (P3) was aware of its possibilities:

*"Another thing that I find intriguing about blockchain is that if you get things through spending a lot of effort, you get rare things, and then you could sell those things for real money. I think that it is a really neat idea. Those guys who built Middle-Earth in Minecraft—they built it for 13 years—I would say that if they could now sell the Middle-Earth as a map, I think that is what they would do."* (P3)

## 5. Discussion

This study examined whether the blockchain adds value to the gaming experience. This was done by creating a blockchain game and demonstrating it to interviewees. This chapter analyzes the results by discussing core findings in the context of relevant literature.

Transparency is one of the core benefits of blockchain-based systems [3,12] and the interviews gave useful feedback on system transparency issues. Many interviewees raised both privacy concerns and possible applications of transparent data. In addition, the traceability of transactions was mentioned by interviewees, which is also emphasized by Casino et al. [8]. We propose researching high-level abstractions, for example, experimental settings where transparency is represented as an independent variable.

Another interview topic was blockchain-related security. The underlying proposal was to determine whether the interviewees trusted blockchain applications. However, not a single participant gave a direct answer to that question. Queiroz and Wamba [11] state that blockchain transparency or trust does not significantly affect blockchain adoption. Based on the findings of this study, blockchain in gaming was not seen as a security threat, although there were some critical replies to transparency and trust issues.

Our findings of player retention are supported partly by previous studies. Drachen et al. [12] argue that games should not be too easy to play to increase retention. According to our findings, some participants stated that the game did not have enough meaningful content to increase their willingness to play it in the future.

Following Drachen et al.'s [12] argument, it would be safe to assume that implementing blockchain into games would not attract individuals to play the game again if the game itself was not perceived as engaging. However, participants who said they would like to try the game again said that the blockchain was the reason, and in more detail, they were interested in creating blockchain assets. Additionally, there were two interviewees who stated that they would want to play the game again, although the game was unengaging for them. This suggests that there is something interesting in the implementation of blockchain rather than in the game itself. The issue of solving player retention can be difficult, as the literature findings show reduced retention for both too-easy and too-difficult games [12,21].

Bouvier et al. [14] presented a model with four types of engagement: environmental engagement, social engagement, self-engagement, and action engagement. The model provided a possibility to analyze the engagement type of those interviewees who identified engaging aspects in the artifact. However, the model by Bouvier et al. [14] defines engagement types on a very conceptual level, while the participants in our interviews identified specific reasons for engagement. That is why we first divided the engagement types into three categories that closely correspond to interview replies: competitive engagement (P4, P6, P9), engaging generation (P5, P8), and engaging progression (P7). From the three categories, engaging generation is the easiest to identify as a blockchain characteristic because the blockchain provides practice for genuine asset ownership. When reflecting on our results of the model by Bouvier et al. [14], the types of engagement would be environmental engagement (curiosity, relaxation, exploration) and self-engagement (customizing and developing a story around the character). Although social engagement did not come up in the answers based on the demonstrated game, the interviews revealed that enabling social engagement (i.e., token trading) would be relevant for a blockchain-based application.

The financial aspects of crypto assets received little attention from the respondents, although blockchains offer room for emerging design solutions for handling monetary issues. Instead, there were varying opinions on what makes assets valuable in the first place (intrinsic versus instrumental values). Regarding exchange, two participants were adamant about the significance of person-to-person trading as opposed to centralized exchange systems. For them, trading is an essential factor in engagement, especially when considering social engagement.

According to our findings, participants were eager to customize assets as soon as they could. It seems that providing genuine asset ownership through blockchain contributes to environmental engagement and self-engagement. The demonstrated game cannot measure how blockchain affects social or action engagement. The social aspect was mentioned several times, which could be one important focus for blockchain game development, for example, enabling token trading.

User-generated content is often mentioned in related literature [3,10]. The interviews included two aspects: modifications and preserving the game. These aspects were primarily seen as a positive possibility. When implementing this feature into games, the developers should consider other related blockchain topics, such as asset valuation and transparency. Adding the possibility for user-generated content directly into the game (e.g., racetracks) seems like a viable choice, according to the interviews and participants' positive views on it.

All the aforementioned key observations and corresponding implications can be seen in Table 6.

**Table 6.** Key observations from interviews and their implications.

| Observations | Implications |
| --- | --- |
| Blockchain technology was seen as a novel technology among gamers. | Using blockchain technology in games can provide a huge competitive advantage, since there are not many game companies using it as their games' core technology. |
| Transparency was perceived as a complex issue. | Even though transparency is one of the fundamental properties of blockchain technology, gamers might not automatically perceive it as a positive element. |
| Asset ownership was perceived as the most important blockchain feature in games. | Asset ownership can potentially increase player engagement and retention. |
| User-generated content was seen as a positive possibility. | Adding the possibility for user-generated content directly into the game seems like a viable choice. |

From a technical standpoint, implementing token trading is straightforward. Since the tokens are already tied to their respective owners, only the function to transfer the tokens between accounts is needed. One way to achieve this is by implementing a token standard. ERC-721 is a commonly accepted standard for unique tokens or NFTs. The standard would also aid interoperability between systems, for example, in common crypto wallets such as MetaMask. For the user, this would emphasize the point that a crypto token is an asset of value that they have genuine ownership of.

A limitation of this study was that only 3 out of the 10 participants could try out the actual game. This was due to the COVID-19 pandemic and meeting restrictions caused by it. However, the artifact was demonstrated via screen sharing. Naturally, in the cases of actual hands-on experience, the participants seemed to be more engaged with the game. There are only a few examples of blockchain games designed for academic purposes [31–33]. Karapapas et al. created a proof-of-concept implementation of a decentralized and fair baseline system for a trading game [31]. However, they did not involve any end-user participants to evaluate their design [31], and neither did Alefs et al. for their designed blockchain gaming platform [32]. Yilmaz et al. investigated the use of non-fungible tokens as trading mechanisms in Virtual Reality Metaverse settings, with only three participants testing their system [33].

Our experiences encourage using qualitative research methods when the study is explorative and inductive. However, measuring player retention and engagement would benefit from more extended experiments and quantitative data gathering. Additionally, questions related to transparency and trust are somewhat challenging to address when emerging technology like the blockchain is being evaluated. The potential for future research on blockchain-based video games is vast. Some recommended research topics emerged from this study:

Design Science Research study on a system that enables user-created content with the blockchain;

The effects of the blockchain on retention based on quantitative game user data;

The effects of the blockchain on different engagement types based on quantitative data by users.

Also, designing an identical game without any blockchain and testing it with a control group could enhance the results of this study.

## 6. Conclusions

The objective of this study was to investigate the effects of blockchain implementation (NFTs and user-generated content), especially on player engagement and retention in games, and how players view blockchain features such as trust and transparency. A unique mobile game utilizing blockchain technology was purposefully designed and created as a university project to demonstrate and evaluate the effects of the blockchain in a game. Qualitative data was gathered via semi-structured interviews with either test play or game demonstration sessions. A total of 10 interviews were conducted, either face-to-face or through online meetings. According to the results, participants' interest in generating blockchain assets or interest in the general implementation of the blockchain seems to resonate with a person's willingness to try the game again. Few distinct reasons for

engagement were found from the analysis, and from those, environmental engagement and self-engagement were most associated with blockchain asset generation.

In conclusion, this study provides initial suggestions on the effects of the blockchain on player retention and engagement. Some of the identified themes are not exclusive features of blockchains, but one of the goals of this study was to learn to look at the blockchain as a tool that can be applied to specific problems whenever they present themselves.

**Author Contributions:** Resources, J.M.; Software, J.N.; Supervision, P.K.; Writing—original draft, I.P., J.N., and P.K.; Writing—review & editing, J.M. All authors have read and agreed to the published version of the manuscript.

**Funding:** This research received no external funding.

**Institutional Review Board Statement:** Not applicable.

**Informed Consent Statement:** Informed consent was obtained from all subjects involved in the study.

**Data Availability Statement:** Not applicable.

**Conflicts of Interest:** The authors declare no conflict of interest.

**Appendix A**

Example interview questions by topic:

1. General questions
   - How would you describe your relationship with video games?
   - How would you differentiate video games from other media entertainment?
   - Why are those aspects important?
   - How do you feel when playing video games?
   - How do you feel about the blockchain? Do you feel the pressure to get involved?
   - What do you have to say about the blockchain game prototype?
   - Would you play the game again? Why/why not?
   - If the blockchain functionalities were implemented in an existing game you play, how would you feel about that?
   - Was the game engaging?
   - How did you feel about the responsiveness in the game?

2. What makes assets valuable
   - Have you ever considered any video game asset/item valuable?
   - How would you differentiate the value of money from that item?
   - Why is that important?
   - Do you think these could be unified?

3. Asset ownership and the lack of a trusted third party
   - In the game you played, how does it make you feel that there isn't a higher authority over the assets than yourself? Explain the feelings.
   - How does the lack of a middleman make you feel from a social point of view?
   - Have you ever considered who actually owns your virtual assets/items on traditional games or software platforms?
   - If so, did it concern you? Why/why not?
   - If so, how would you differentiate the concern from concern over losing something you physically own?

4. Data on blockchain, immutable and transparent
   - All of the transactions done in the game are public, and your racers can be identified using your address. How do you feel about that?
   - Do you think there's value in the possibility of multiplayer interactivity after developer support ends? Why?
   - How about 3rd party modding possibilities during development? Why?

5. Security and trust on the blockchain
   - How would you describe your security concerns regarding information systems in general?
   - From what you've experienced, would you say that you trust the blockchain? Why/why not?

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
