# Peer review of "Users’ Perceptions of Key Blockchain Features in Games"

_futureinternet, doi:10.3390/fi14110321_

Round 1

Reviewer 1 Report

Dear Authors.
Thanks for submitting your paper to Future Internet, an MDPI Journal with a cite score of 5.4 and Tracked for Impact Factor.
The article you submitted concerns blockchain applied in the Gaming sector, which I agree is an overlooked aspect of the blockchain literature. You conducted a qualitative study based on Interviews.
I must say that the paper is among the few with the “gift” of being clearly readable and with good English. On this aspect, my opinion is 100% positive.
However, the scientific content is a bit tentative and requires more elaboration. The article is more adapt for a conference than a Journal at the moment.

Please accept some comments and suggestions.

1)     I am very dubious about the validity of the dataset. You propose 10 participants, of which only three have played the game, and others watched a video. I don’t want to be rude, but I believe that the opinion of 10 people is irrelevant to this or any other study. I have read studies with a low number of interviewed people, but those people had a high understanding and impact. To make an example, there are famous gamers whose knowledge and experience are undisputed. If you presented a study involving 10 of the most famous online gamers, then it would have been valuable. But 10 random people are not. How are those people selected? Who are they? Are they involved in some specific and undisputed knowledge in the sector? If not, I am afraid that their number is too low to bear any significance.

2)     There is not a clear link between theory and results in this study. There is an introduction on assets and a distinction of tokens (that, however, it is not really clear where it comes from and why it is chosen), but they seems to be no finality in what the study is trying to highlight. The background should build a framework, even visible such as a scheme that helps the reader understand and interpret the data reported in the study. In principle, the paper is super interesting, but in the complex, it lacks some fundamental pillars of the qualitative study structure.

3)     What are the questions? We have the research questions in the introduction (which are unfortunately not adequately addressed in the discussion section), but we don’t have the queries made to the participant. What are the participants responding to? Is there a difference between those who played and those who didn’t? Please provide some samples in the appendix. At line 272 there  is “engagement” two times, is there something else supposed to be in place of the last “engagement”?

4)     Did you get something out of the coding? Which coding type have you utilized? What is the source? There is a considerable confusion in the results paragraph because there is no guide to help the reader understand what is analyzed and why. What is the succession of content, and what literature scheme is followed.

5)     Table 2 is good. It shows a breakdown of choices. But this should be just one of tons of tables such as these. Probably at least one for any aspect of the study you are investigating.

6)     Finally, the discussion and conclusion do not really highlight this study's value. A table should summarize the main findings putting in relation: the topic, the background study citation, a piece of a related sentence, and the final intuition. That would also help other researchers that may undertake reviews on this topic. Also, it should be clearly highlighted how academic literature can benefit from this study, and what is the step forward that has been made.

7)     Lastly, the title is in caps. I believe it should be just the first letter.

Again, sorry for being harsh, but I believe that this study has some potential, and it would be a shame to have it published in this state.
Good luck with your research!

Author Response

Dear Editor and reviewers, 

Thank you for insightful comments. We detail our responses in the text below, and hopefully the changes meet all the requests. We believe that the article has improved significantly in this iteration, and we are looking forward to receive your feedback. 

Reviewer 1 

1)     I am very dubious about the validity of the dataset. You propose 10 participants, of which only three have played the game, and others watched a video. I don’t want to be rude, but I believe that the opinion of 10 people is irrelevant to this or any other study. I have read studies with a low number of interviewed people, but those people had a high understanding and impact. To make an example, there are famous gamers whose knowledge and experience are undisputed. If you presented a study involving 10 of the most famous online gamers, then it would have been valuable. But 10 random people are not. How are those people selected? Who are they? Are they involved in some specific and undisputed knowledge in the sector? If not, I am afraid that their number is too low to bear any significance. 

Thank you for the critical viewpoint, and it is true that having more participants in the interviews would make this article stronger. The IkuneRacer game was made for research purposes, and unfortunately it is not operative anymore. Therefore, we cannot conduct any additional interviews based on this IT artifact. 

However, we must underline that there are only a few examples of blockchain games or platforms designed for academic purposes. Those that we were able to find neither Karapapas et al. (2022) or Alefs et al. (2022) did not involve any end user participants to evaluate their design, and Yilmaz et al. (2022) had only three. 

Our aim was to find perceptions of ordinary gamers and get their perceptions of the potential of blockchain technology. The nature of this study is exploratory, and therefore our goal has not been to validate any hypotheses. Some of the interviewees were recruited based on their participation in a relevant university course, while others were recruited through personal connections. Three of the interviewees were studying in or had a degree from a university of applied sciences, and the other seven were studying in or had a degree from a university. We added this information about recruitment and participants to the manuscript. 

2) There is not a clear link between theory and results in this study. There is an introduction on assets and a distinction of tokens (that, however, it is not really clear where it comes from and why it is chosen), but they seems to be no finality in what the study is trying to highlight. The background should build a framework, even visible such as a scheme that helps the reader understand and interpret the data reported in the study. In principle, the paper is super interesting, but in the complex, it lacks some fundamental pillars of the qualitative study structure. 

We have improved the clarity of background and discussion sections, which aims to make better connection between the theory and results. We have added a table in the discussion section to highlight key findings of this study. We interpret that the reviewer is leaning his/her ideas to methods that are building theories or frameworks such as the Grounded Theory method. As said, this study’s approach is exploratory. 

3) What are the questions? We have the research questions in the introduction (which are unfortunately not adequately addressed in the discussion section), but we don’t have the queries made to the participant. What are the participants responding to? Is there a difference between those who played and those who didn’t? Please provide some samples in the appendix. At line 272 there  is “engagement” two times, is there something else supposed to be in place of the last “engagement”? 

Questions has been added as Appendix 1. We also added a table in the section 3.2 that indicates which of the participants were able to play the game, and which were not. 

The extra ‘engagemement’ word removed. 

4)     Did you get something out of the coding? Which coding type have you utilized? What is the source? There is a considerable confusion in the results paragraph because there is no guide to help the reader understand what is analyzed and why. What is the succession of content, and what literature scheme is followed. 

We have added couple of paragraphs about thematic coding and descriptions of laddering used when conducting interviews (Terry et al. 2017). 

5) Table 2 is good. It shows a breakdown of choices. But this should be just one of tons of tables such as these. Probably at least one for any aspect of the study you are investigating. 

We have added three tables, and currently there is a table in each chapters excluding introduction and conclusion. 

6)     Finally, the discussion and conclusion do not really highlight this study's value. A table should summarize the main findings putting in relation: the topic, the background study citation, a piece of a related sentence, and the final intuition. That would also help other researchers that may undertake reviews on this topic. Also, it should be clearly highlighted how academic literature can benefit from this study, and what is the step forward that has been made. 

A new table presenting key observations and their implications was included. 

7)     Lastly, the title is in caps. I believe it should be just the first letter. 

Corrected. Thank you. 

Reviewer 2 Report

The idea of the paper is interesting, however, it needs substantial improvement before publication. Below are the comments that need to be addressed

1.     The technical content is good however the motivation of the paper is not clear. Why is your proposal needed? What are the challenges involved? What solutions already exist for the problem you want to solve? What are their limitations and drawbacks?

2.     Abstract need to be revised, results should be added in abstract

3.     Add some latest references

4. All cited papers should be in proper order.

5.     In the Conclusion section, authors can add 1-2 good future directions.

6.     May add some graphs in the result section to improve it

7.     There is a need for detail of experimental results, how interview validity can be ensured

8.     The English writing should be improved by native speaker.

9.     The conclusion part might be revised underlining the novelty of the approach.

Author Response

Dear Editor and reviewers, 

Thank you for insightful comments. We detail our responses in the text below, and hopefully the changes meet all the requests. We believe that the article has improved significantly in this iteration, and we are looking forward to receive your feedback. 

REVIEWER 2 

  1. 8. The technical content is good however the motivation of the paper is not clear. Why is your proposal needed? What are the challenges involved? What solutions already exist for the problem you want to solve? What are their limitations and drawbacks?

Blockchain is an emerging technology that has the potential to revolutionize the gaming industry among a wide range of different business fields. So far, only a few studies have been conducted about blockchain gaming. We have added several references to articles published during this year that have made designs for blockchain games or platforms. Limitations of this study have been rewritten and moved to Discussion section, where our study’s drawbacks are hopefully more clearly presented. 

  1. Abstract need to be revised, results should be added in abstract 

More results are included in the abstract.  

  1. 10. Add some latest references 

More recent references have been added.  

  1. 11. All cited papers should be in proper order.

We have done thorough check of our citations, and we cannot locate any disorder that the reviewer is referring to. 

12. In the Conclusion section, authors can add 1-2 good future directions.  

We have made changes to conclusion section and stated several suggestions for future studies. 

  1. 13.     May add some graphs in the result section to improve it 

We have added three tables, and currently there are a table in each chapter from 2 to 6.  

  1. 14.     There is a need for detail of experimental results, how interview validity can be ensured 

Several paragraphs of details about experiment method added. 

  1. The English writing should be improved by native speaker.

We have done a final proof reading of this text and improved its quality. As a reminder, the first reviewer thanked the quality of the language of this manuscript.  

  1. 16. The conclusion part might be revised underlining the novelty of the approach.

Conclusion section has been revised. 

Reviewer 3 Report

1.This article lacks innovation: there have been quite a few researches on mobile games based on blockchain and smart contracts, such as CryptoKitties that is mentioned in the article. The games are designed for too few users to reflect the benefits of blockchain that can enhance the playability of the game.(For example, creative workshops shared by players whose intellectual property rights are not in question)

2. In the section of validation, the number of the interviewee is to small,thus resulting in inaccurate conclusions of statistics. Also subjective factors should be avoided in questionaire design. For example, according to the answer of one interviewee in page8,Line318 : I would play a game like that, but of course not with cars because I think cars are the most boring thing.

3.The experimental results are not straightforward.

a. page8,Line334,Table 2 and page11, Line456-457: subjective answers of interviewees are used to explain retention and engagement of the game, but no follow-up observation is included, which is necessary for complete evaluations.

b. The experiment did not include a control group for fair comparisons, e.g. by designing a basically identical game, but without blockchain technology and smart contract, and for a comparative analysis of subjects.

c. page10,Line451-452: the authors considered the influence of whether the content of the game itself is interesting on the research conclusion, but no experiments were designed to exclude this interference.

4. The subtitles of chapters 2 and 4 are incorrectly labeled.

Author Response

Dear Editor and reviewers, 

Thank you for insightful comments. We detail our responses in the text below, and hopefully the changes meet all the requests. We believe that the article has improved significantly in this iteration, and we are looking forward to receive your feedback. 

REVIEWER 3 

  1. This article lacks innovation: there have been quite a few researches on mobile games based on blockchain and smart contracts, such as CryptoKitties that is mentioned in the article. The games are designed for too few users to reflect the benefits of blockchain that can enhance the playability of the game.(For example, creative workshops shared by players whose intellectual property rights are not in question)

If the reviewer is referring to the small number of participants in this study, we have addressed this issue in the review #1. If the reviewer is arguing that there are only few blockchain related games in the market, and they are designed only to a narrow segment of players, then we believe that this in fact indicates how vast potential our research theme has. Creative workshops could be very interesting way to design next generation blockchain games. 

  1. In the section of validation, the number of the interviewee is to small,thus resulting in inaccurate conclusions of statistics. Also subjective factors should be avoided in questionaire design. For example, according to the answer of one interviewee in page8,Line318 : I would play a game like that, but of course not with cars because I think cars are the most boring thing.

This is a qualitative study that aims to gather as wide variety of different perceptions of key blockchain features as possible. Subjective factors are highly relevant. 

19.The experimental results are not straightforward. 

  1. page8,Line334,Table 2 and page11, Line456-457: subjective answers of interviewees are used to explain retention and engagement of the game, but no follow-up observation is included, which is necessary for complete evaluations.

This is an explorative study, which do not aim to validate or verify results. 

  1. The experiment did not include a control group for fair comparisons, e.g. by designing a basically identical game, but without blockchain technology and smart contract, and for a comparative analysis of subjects.

Again, this study’s method was qualitative. Based on the findings of this study, it is easier to see what elements of blockchain technology are perceived relevant by participants. 

  1. page10,Line451-452: the authors considered the influence of whether the content of the game itself is interesting on the research conclusion, but no experiments were designed to exclude this interference.

Content is an elemental part of any game design. However, if we had been able to design an abstract game such as Tetris, this could have been perhaps solvable. Nevertheless, as a positive aspect, using racing cars as the main characters was also beneficial, since it is familiar theme to all participants. 

  1. The subtitles of chapters 2 and 4 are incorrectly labeled.

Corrected. 

Round 2

Reviewer 1 Report

Dear Authors.
I see the study has improved. Just to notice, when you say "Those that we were able to find neither Karapapas et al. (2022) or Alefs et al. (2022) did not involve any end user participants to evaluate their design, and Yilmaz et al. (2022) had only three." it does not justify to produce other research like this. As you know a lot of articles online are rubbish but that does not justify the production of other rubbish. In any case, it it is not proving or stating anything but is just perceived as a viewpoint it can be acceptable. 
Good luck with your research. 

Author Response

Thank you for you time and feed back.
We made the following changes:

Into the beginning of chapter 3:

This study follows exploratory research paradigm (e.g., Robson, 2002), which is described as a mapping research of little-known areas of study. It aims to discover the nature of the problem and guide academics to understand it better. Exploratory research is flexible, and it can use both quantitative and qualitative methods. However, exploratory research does not aim to test hypothesis based on existing theories and knowledge.

Into the abstract (along with other minor changes) :  This study follows exploratory research paradigm, which aims to map research of little-known areas.

A new reference is added :

Robson, C. (2002). Real world research: A resource for social scientists and practitioner-researchers. (Vol. 2): Blackwell Oxford.

Reviewer 2 Report

The authors have addressed all my comments, i have no further comment.

Author Response

Thank you.

Reviewer 3 Report

The explanations provided are reasonable. Though only as an explorative study, this paper can be considered for publication after minor improvement.

To be specific, I suggest that the claim of "This is an explorative study, which do not aim to validate or verify results. " should be included somewhere in the paper (e.g. keywords, abstract), which can align with readers' expectation. 

Author Response

(The authors gave the same response as above.)
